# Transfer Entropy of West Pacific Earthquakes to Inner Van Allen Belt Electron Bursts

**DOI:** 10.3390/e24030359

**Published:** 2022-03-02

**Authors:** Cristiano Fidani

**Affiliations:** Central Italy Electromagnetic Network, 63847 San Procolo, FM, Italy; c.fidani@virgilio.it

**Keywords:** strong earthquakes, electron precipitations, statistical correlation, mutual information, transfer entropy

## Abstract

Lithosphere-ionosphere non-linear interactions create a complex system where links between different phenomena can remain hidden. The statistical correlation between West Pacific strong earthquakes and high-energy electron bursts escaping trapped conditions was demonstrated in past works. Here, it is investigated from the point of view of information. Starting from the conditional probability statistical model, which was deduced from the correlation, the Shannon entropy, the joint entropy, and the conditional entropy are calculated. Time-delayed mutual information and transfer entropy have also been calculated analytically here for binary events: by including correlations between consecutive earthquake events, and between consecutive earthquakes and electron bursts. These quantities have been evaluated for the complex dynamical system of lithosphere-ionosphere; although the expressions calculated by probabilities resulted in being valid for each pair of binary events. Peaks occurred for the same time delay as in the correlations, Δt = 1.5–3.5 h, and as well as for a new time delay, Δt = −58.5–−56.5 h, for the transfer entropy; this last is linked to EQ self-correlations from the analysis. Even if the low number of self-correlated EQs makes this second peak insignificant in this case, it is of interest to separate the non-linear contribution of the transfer entropy of binary events in the study of a complex system.

## 1. Introduction

Numerous experiments have followed the discovery of the Van Allen belts [1] to determine safe conditions for near-Earth space exploration [2]. Given that sudden fluxes of high-energy particles in the South Atlantic Anomaly region constitute a significant hazard for satellite outages [3], they have been thoroughly studied. Afterwards, sudden variations in high-energy charged particle fluxes near the South Atlantic Anomaly were for the first time associated with seismic activity by Voronov et al. [4]. Subsequently, flux variations of charged particles associated with strong earthquakes (EQs) were detected using the Intercosmos-24 satellite [5,6]. These variations were described as precipitating particles that escaped the trapped conditions of the geomagnetic field. Such sudden fluxes have been statistically analyzed in relation to the seismic activity in various near-Earth space experiments, such as the MIR orbital station, the METEOR-3, GAMMA, and SAMPEX satellites [7]. These fluxes have shown particle burst correlations with 2–5 h ahead on strong EQs. Moreso, the SAMPEX database has been studied, revealing a 3–4 h correlation with precipitating high-energy electrons, anticipating strong shallow mainshocks [8]. All of these studies considered up to only a few months of recordings. The NOAA-15 satellite high-energy particle database, collecting more than 16 years of counting rates, has been studied: electron burst (EB) events resulted at 2–3 h in advance of strong EQ events [9]. This correlation resulted as statistically significant for mainshocks of M > 6 striking the Indonesian and Philippine areas, occurring at a depth of less than 200 km. The possible disturbances above the EQ epicenters due to particle drift have also been estimated to be in the range of 4–6.5 h before strong seismic events, suggesting a possible causal connection [10]. A conditional probability formulation of EQ occurrence following satellite measurements, considered as binary events, prognosticated an increase in EQ forecasting probability [11]. In a recent publication, results of a possible EQ forecasting experiment by NOAA satellites were estimated, as well as the number of false alarms. The complete equivalence of this formulation with those of the probability gain for a test verification was presented [12]. The hypothesis of magnetic pulses able to push electrons out of the trapped conditions was quantitatively evaluated, starting from recent results of magnetic measurements [13]. Finally, a probability gain has been estimated for dependent events. In fact, considering the probability gain from EBs and the probability gain from magnetic pulses, a combined probability gain was calculated supposing to use both together. Thus, the probability gains due to one of two types of events indifferently retrieved in the same space-time region, was shown to be greater than each EB and magnetic pulse probability gains [12].

The approach of the information theory is able to detect causal information flows in complex systems [14]. Complex system science is concerned with the study of systems that are composed of many simple elements interacting in a non-trivial fashion. The non-triviality of the interaction occurs when the system characterized by many components, which, if they interacted trivially, would be the domain of statistical mechanics, has well-defined and constrained interactions [15], which lead to non-linear dynamics. The complex dynamics of the Earth’s tectonic plates manifest as self-similarity and an absence of characteristic length-scale in magnitude, space, and time of EQs [16]. A complexity measure was proposed as a change of entropy in natural time under time reversal [17], the study of which enables the determination of the occurrence time of major EQs in Japan [18]. Furthermore, it enables the estimation of EQ epicentral regions [19] when the analysis of seismicity is made in natural time [20,21]. The fluctuations of entropy change have been studied in an interval of seismic records in Mexico, showing an evident increase months before the strongest EQ of the last century [22]. Information entropy has been used in seismology as an indicator of the evolution of a system [23]. Entropy calculated on the statistical distribution of EQs throughout the world from 1963 to 2012 suggested that it may represent an interrelation in seismic data [24]. Mutual information was also calculated and adopted as a measure of similarity between regions, providing an intuitive and useful representation of the complex relationships among EQs [25].

Complex system science focuses on emergent global behavior or organization from non-trivial distributed interactions, linking with the concepts of self-organization [26]. The lower ionosphere has been recently shown to be capable of reaching a critical state several days before and after the mainshock [27], thus highly suggesting it to be a complex system. Thermal and electromagnetic anomalies in the Earth’s atmosphere, ionosphere, and magnetosphere have been investigated in an open complex system with dissipation, with the LAIC model [28]. More recently, an algorithm has been presented to detect EQ preparation processes where different levels, starting from underground through the atmosphere and up to the ionosphere and magnetosphere, must be finely sampled [29]. However, information theory has not yet been utilized in the lithosphere to ionosphere system, even if non-linear processes are involved. Thus, mutual information and transfer entropy have been in this study, for the first time, calculated in the context of precipitating EBs from the inner Van Allen belt, which preceded strong EQs. After resuming the main steps to obtain the correlation between EQs with M ≥ 6 in the West Pacific and the high-energy EBs detected by the NOAA-15 satellite, the information-theoretic approach was adopted for the lithosphere–ionosphere system in Section 2. The analytical calculations of entropy, delayed mutual information, and the conditional information transfer for the general case of a couple of binary events are reported in Section 3. Section 4 discusses the main results obtained in the previous section. Conclusions are summarized in Section 5.

## 2. Materials and Methods

This study was based upon the large data set of particle counting rates, EQs, and geomagnetic indexes that are deposited in publicly available databases. A well-established method [9,10,11,12] permits to define EBs from the 30–100 keV electron counting rate database during geomagnetically quiet days defined by geomagnetic indexes. Then, EBs are correlated to EQs running on 16.5 years of data by filling histograms for intervals of positive and negative delays between each other. For electrons, the medium energy proton and electron detector onboard the NOAA-15 polar satellite [30] collected counting rates from 1998: http://www.ngdc.noaa.gov/stp/satellite/poes/dataaccess.html (accessed on 28 February 2022). Being that the electron counting rates had been contaminated by protons [31] from the lower energy range, corrections were carried out using software downloaded from the Virtual Radiation Belt Observatory: http://virbo.org/POES#Processing (accessed on 28 February 2022). Regarding the geomagnetic field, it was re-evaluated together with L-shells on the NOAA-15 orbit using the International Geomagnetic Reference Field (IGRF-12) model [32], downloaded at: http://www.ngdc.noaa.gov/IAGA/vmod/igrf.html (accessed on 28 February 2022). Geomagnetic Ap indexes and Dst variations were downloaded at the links https://www.ngdc.noaa.gov/geomag/data.shtml (accessed on 28 February 2022) and http://wdc.kugi.kyoto-u.ac.jp/dst_final/index.html (accessed on 28 February 2022), respectively. The escaping conditions of trapped electrons were also determined with respect to the geomagnetic field by calculating their minimum mirror point altitudes through the UNILIB libraries [33], which were downloaded at https://www.mag-unilib.eu (accessed on 28 February 2022). Finally, EQ events were downloaded at https://earthquake.usgs.gov/earthquakes/search/ (accessed on 28 February 2022), and the database was declustered using CLUSTER 2000 software [34], downloadable at https://www.usgs.gov/media/images/cluster2000 (accessed on 28 February 2022).

The only electron flux detected by the NOAA-15, which was the difference between the channel 30 keV–2.5 MeV and the channel 100 keV–2.5 MeV coming from the zenith, gave positive results for a correlation with strong EQs [9]. Electron fluxes of greater energy and coming from other directions, as well as any proton fluxes detected by NOAA, produced no detectable correlations with strong EQs. After having been corrected for proton contamination, electrons were verified to have mirror altitudes less than 100 km along the drift period, which meant being disturbed by the trapped conditions in the inner Van Allen belts. The dynamics of electrons were described using adiabatic invariants retrieved using geomagnetic quantities and represented in a 4-dimensional matrix of time, L-shell, pitch angle, and geomagnetic field. As electron loss is principally induced by solar activity, time intervals when Ap and Dst indexes overcame appropriate thresholds were excluded. Thus, an electron counting rate (ecr) fluctuation was of a statistical origin with less than 1% probability, when the condition Poisson(ecr) < 0.01 was satisfied. This was considered as an EB influenced by the Earth’s surface. Furthermore, EBs detected along the same semi-orbit were considered as one EB. A correlation between EQs and EBs has been recently calculated so as to unambiguously define EBs. It was obtained by including only an EB L-shell in a restricted interval [12]. A peak of correlation appeared when considering EQs at a depth of less than 200 km, located in a large portion of the Earth’s crust in the West Pacific, with a minimum magnitude of 6. The peak occurred at the Δt = 1.5−3.5 h interval. Being the conditional probability distribution of an EQ occurrence given the observation of an EB obtained by:P(EQ|EB) = G_Δt_ P(EQ) = P(EQ) + corr(EQ,EB){P(EQ)[1 − P(EQ)][1 − P(EB)]/P(EB)}^½^(1)
The corresponding peak in conditional probability (1) of the correlation peak is shown in Figure 1. It deviates significantly from the average probability of an EQ, the unconditional P(EQ).

Information-theoretic measures are based on probability distributions; hence, they can be calculated independently from the model and are expected to be capable of capturing non-linear interactions [35]. The information in the information theory is quantified by Shannon entropy [14], and for a binary system, is defined as:H(E) = −Σ_{0,1}_ P(x) log_2_ P(x)(2)

The logarithm of base 2 was chosen because binary events E were considered, and the unit of information was in bits. This measures the uncertainty associated with E, and H(E) is maximum if 0 and 1 are both equally probable for E. Therefore, H(E) is zero when only 0 or 1 is more likely to occur.
H(E_1_,E_2_) = −ΣΣ_{0,1}_ P(x,y) log_2_ P(x,y)(3)
is the joint entropy [14], and P(x,y) is the joint probability distribution. This describes the amount of uncertainty that characterizes both sets of events, E_1_ and E_2_. On the other hand, the conditional entropy H(E_1_|E_2_) [14]
H(E_1_|E_2_) = −ΣΣ_{0,1}_ P(x,y) log_2_ P(x|y)(4)
measures the uncertainty about the event E_1,_ which remains after the other event E_2_ is manifested. In analogy, P(x|y) is the conditional probability. H(E_1_|E_2_) = H(E_1_) when the knowledge of the event E_2_ does not reduce the uncertainty on the event E_1_. Moreover, in the case of dependency between events, each one contains information on the other. To quantify the amount of information that the event E_2_ has on E_1_, mutual information [14] is defined as:I(E_1_;E_2_) =ΣΣ_{0,1}_ P(x,y) log_2_ {P(x,y)/[P(x)P(y)]}(5)
which is a symmetric distribution with I(E_1_;E_2_) = I(E_2_;E_1_), which is not able to detect the direction of the information flux between the events. An asymmetric measure, named time-delayed mutual information [36], can be obtained if an investigator is interested in causality, by introducing a time-lag parameter Δt between events E_1_ and E_2_ as:I(E_1_;E_2 +Δt_) = ΣΣ_{0,1}_ P(x,y_+Δt_) log_2_ {P(x,y_+Δt_)/[P(x)P(y_+Δt_)]}(6)

If Δt = t_x_ − t_y_, the sign of Δt can be used to infer the direction of influence [14] when relation (6) exhibits a peak in Δt. A negative Δt for a peak in mutual information indicates that E_1_ shares a maximum amount of information with the future state of event E_2_. Consequently, it is possible to affirm that E_1_ drives E_2_. Instead, when Δt is positive, this indicates that E_1_ shares a maximum amount of information with the past of E_2_, which is equivalent to affirming that E_1_ is driven by E_2_ [37]. Time-delayed mutual information works also when non-linear links characterize interactions between two event time series [38]. While shared history and common external driving effects between two processes seem to escape from identification by means of mutual information [39], Schreiber [40] pointed out that a condition on the past state of E_2_ must be fixed in Equation (6) using conditional mutual information. Then, transfer entropy is defined as a conditional time-delayed mutual information [41] from the event E_2_ to E_1_ with respect to a time lag Δt.
TE_E2->E1_ = I(E_1+1_;E_1_|E_2+Δt_) = ΣΣΣ_{0,1}_ P(x_1+1_,y_1_,z_2+Δt_) log_2_ [P(x_1+1_|y_1_,z_2+Δt_)/P(x_1+1_|y_1_)](7)

Transfer entropy from E_2_ to E_1_ is based on the past of both E_1_ and E_2_, and quantifies how reliably the event E_2_ can predict the occurrence of the event E_1_.

## 3. Results

Considering the binary events EQ and EB, the sum (2) has two terms that represent the presence or absence of the event. Being so, the Shannon entropies are:H(EQ) = −P(EQ) log_2_ P(EQ) − [1 − P(EQ)] log_2_ [1 − P(EQ)](8)
and
H(EB) = −P(EB) log_2_ P(EB) − [1 − P(EB)] log_2_ [1 − P(EB)](9)

These can be evaluated immediately because the unconditioned probabilities are known for the singularly considered systems of West Pacific EQs and precipitating EBs from the internal Van Allen belts [12]. The total entropy of the two systems is the sum of their entropies. While, the entropy of the combined system is the joint entropy:H(EQ,EB) = −P(EQ|EB)P(EB) log2 [P(EQ|EB)P(EB)] +           −[1 − P(EQ|EB)]P(EB) log2 {[1 − P(EQ|EB)]P(EB)} +                      −[P(EQ) − P(EQ|EB)P(EB)] log2 [P(EQ) − P(EQ|EB)P(EB)] +                     −{1 − P(EB) − P(EQ)[1 −P(EQ|EB)]} log2 {1 − P(EB) − P(EQ)[1 − P(EQ|EB)]}(10)
where Σ_EB = {0,1}_ P(EQ,EB) = P(EQ) and Σ_EQ = {0,1}_ P(EQ|EB) = 1, being P(EQ = 1) = P(EQ) and P(EQ = 0) = 1 − P(EQ), were used. The origin of the rows reported in (10) can be clarified in a Venn diagram (see Figure 2 left).

Relation (10) can be rewritten as:H(EQ,EB) = H(EB) + H(EQ|EB)(11)
where the relation (9), the logarithm properties and the definition of conditional entropy are taken into account. Thus, introducing the probability gain G_Δt_ = P(EQ|EB)/P(EQ) by (1) with a time delay Δt = T_EQ_ – T_EB_ of correlations [12] between EQs at the time T_EQ_ and EBs at the time T_EB_, the conditional entropy can be rewritten in a more synthetic form as:H(EQ|EB) = G_Δt_ P(EQ)P(EB) log_2_ {[1 − G_Δt_ P(EQ)][1 − G_Δt_ P(EB)]/[G_Δt_ − G_Δt_ P(EB)]} +        −P(EQ) log_2_ {P(EQ)[1 − G_Δt_ P(EB)]/[1 − P(EB)]} − P(EB) log_2_ [1 − G_Δt_ P(EQ)] +            −[1 − P(EB) − P(EQ) + G_Δt_ P(EQ)P(EB)] log_2_ {1 − P(EQ)[1 − G_Δt_ P(EB)]/[1 − P(EB)]}.(12)

The joint entropy is a symmetric function where H(EB,EQ) = H(EQ,EB), while the conditional entropy follows the Bayes rule [14]:H(EB|EQ) = H(EQ|EB) − H(EQ) + H(EB)(13)

Relations among the entropies are represented in Figure 2 right, by a Venn diagram. The conditional entropy can be also calculated for a time delay Δt, as well as for the joint entropy. Starting from (1) for EQs in West Pacific and EBs detected by NOAA-15, the entropies are reported in Figure 3 for the same range of delays as in Figure 1. In this regard, it must be remembered that N_EQ_ = 600, on page 11 of the reference [12] is the number of EQs with M ≥ 6 that struck the West Pacific during the entire 16.5-year period. However, the number of EQs that occurred during the 6953 pairs of hours would only be considered, which is N_EQ_ = 57, and P(EQ|EB) = 0.0081 EQs for each pair of hours. Values of the alarm rate and the failure rate would be consequently corrected in [12] to 0.77 and 0.23, respectively. A N_EB_ = 1892 EBs was considered in the same time interval, so that P(EB) = 0.27. Joint and conditional entropies were retrieved by (10) and (12) according to the time delay Δt, both of which showed a minimum for Δt = 1.5–3.5 h corresponding to the correlation peak.

The time delay Δt can be also be introduced for the mutual information, obtaining the time-delayed mutual information (6), it can be rewritten using G_Δt_ as: I(EQ;EB) = −G_Δt_ P(EQ)P(EB) log_2_ {[1/ G_Δt_ − P(EQ)][1 − G_Δt_ P(EB)]} − log_2_ {[1 − P(EQ)][1 − P(EB)]} +     + P(EQ) log_2_ {[1 − P(EQ)][1 − G_Δt_ P(EB)]} + P(EB) log_2_ {[1 − P(EB)][1 − G_Δt_ P(EQ)]} + + {1 − P(EB) − P(EQ)[1 − G_Δt_ P(EB)]} log_2_ {1 − P(EB) − P(EQ)[1 − G_Δt_ P(EB)]},(14)
which can be connected to the entropies
I(EQ;EB) = H(EQ) + H(EB) − H(EQ,EB)(15)
and
I(EQ;EB) = H(EQ) − H(EQ|EB)(16)
using the chain rule for the entropies. Both (15) and (16) suggest that I(EQ;EB) = 0 when EQ and EB are independent events. Relations among the entropies and the mutual information are represented in Figure 2 right, by a Venn diagram. The plot of the relation (14) is reported in Figure 4 together with H(EQ|EB) and H(EQ), with respect to the time delay Δt. The time-delayed mutual information reveals a maximum for Δt = 1.5–3.5 h, in analogy with the entropies.

Finally, the transfer entropy (TE) can also be calculated analytically, starting from the conditional probability distributions. In fact, the eight terms of TE_EB->EQ_ contain the joint probabilities P(EQ ∩ EQ_+1_ ∩ EB_+Δt_) = P(EQ ∩ EQ_+1_|EB_+Δt_)P(EB_+Δt_), where +1 indicates the time step of 2 h ahead, and Δt indicates the general time ahead of 2-h n-steps. Note that the time zero for the West Pacific to NOAA system is the interval from −0.5 to 1.5 h, and +1 is from 1.5 to 3.5 h, which coincides with the time for the statistical correlation. From (1), considering P(EQ_+1_) = P(EQ), and P(EB_+Δt_) = P(EB),
P(EQ ∩ EQ_+1_|EB_+Δt_) = P(EQ ∩ EQ_+1_)G_Δt_(EQ ∩ EQ_+1_)(17)
with
G_Δt_(EQ ∩ EQ_+1_) = 1 + corr(EQ ∩ EQ_+1_,EB_+Δt_){[1/P(EQ ∩ EQ_+1_) − 1][1/P(EB) − 1]}^½^(18)
and still from (1)
P(EQ ∩ EQ_+1_) = P(EQ)^2^ + corr(EQ,EQ_+1_)[1 − P(EQ)](19)

Now alternating the events EQ, EQ_+1_, and EB_+__Δt_ from 0 to 1, and considering the sums Σ_EQ = {0,1}_ P(EQ,EQ_+1_,EB) = P(EQ_+1_,EB), Σ_EQ+1 = {0,1}_ P(EQ,EQ_+1_,EB) = P(EQ,EB), and Σ_EB = {0,1}_ P(EQ,EQ_+1_,EB) = P(EQ,EQ_+1_), the eight joint probabilities can be written as:P(0,0,0) = P(EQ ∩ EB_+Δt_) + P(EQ_+1_ ∩ EB_+Δt_) + P(EQ ∩ EQ_+1_)[1 − G_Δt_(EQ ∩ EQ_+1_)P(EB)] + 1 − 2P(EQ) − P(EB)(20)
P(0,0,1) = −P(EQ ∩ EB_+Δt_) − P(EQ_+1_ ∩ EB_+Δt_) + P(EB)[1 + G_Δt_(EQ ∩ EQ_+1_)P(EQ ∩ EQ_+1_)](21)
P(0,1,0) = −P(EQ_+1_ ∩ EB_+Δt_) − P(EQ ∩ EQ_+1_)[1 − G_Δt_(EQ ∩ EQ_+1_)P(EB)] + P(EQ)(22)
P(1,0,0) = −P(EQ ∩ EB_+Δt_) − P(EQ ∩ EQ_+1_)[1 − G_Δt_(EQ ∩ EQ_+1_)P(EB)] + P(EQ)(23)
P(0,1,1) = P(EQ_+1_ ∩ EB_+Δt_) − G_Δt_(EQ ∩ EQ_+1_)P(EQ ∩ EQ_+1_)P(EB)(24)
P(1,0,1) = P(EQ ∩ EB_+Δt_) − G_Δt_(EQ ∩ EQ_+1_)P(EQ ∩ EQ_+1_)P(EB)(25)
P(1,1,0) = P(EQ ∩ EQ_+1_)[1 − G_Δt_(EQ ∩ EQ_+1_)P(EB)](26)
and P(1,1,1) can be directly retrieved from (17). For what concern the log_2_ arguments, from the Bayes theorem and with
P(EQ ∩ EQ_+1_ ∩ EB_+Δt_)P(EQ_+1_)/[P(EQ ∩ EQ_+1_)P(EQ_+1_ ∩ EB_+Δt_)] = P(EQ ∩ EQ_+1_|EB_+Δt_)P(EB_+Δt_|EQ_+1_)(27)
after a lot of algebra, they can be presented as:L(0,0,0) = [1 − P(EQ)]P(0,0,0)/{[1 − 2P(EQ) − P(EQ ∩ EQ_+1_)][1 − P(EQ) − P(EB)[1 − G_Δt+1_P(EQ)]]}(28)
L(0,0,1) = [1 − P(EQ)]P(0,0,1)/{[1 − 2P(EQ) − P(EQ ∩ EQ_+1_)][1 − G_Δt+1_P(EQ)]P(EB)}(29)
L(0,1,0) = P(0,1,0)/{[P(EQ) − P(EQ ∩ EQ_+1_)][1 − G_Δt+1_P(EB)]}(30)
L(1,0,0) = [1 − P(EQ)]P(1,0,0)/{[P(EQ) − P(EQ ∩ EQ_+1_)][1 − P(EQ) − P(EB)[1 − G_Δt+1_P(EQ)]]}(31)
L(0,1,1) = P(0,1,1)/{G_Δt+1_P(EB)[P(EQ) − P(EQ ∩ EQ_+1_)]}(32)
L(1,0,1) = [1 − P(EQ)]P(1,0,1)/{P(EB)[1 − G_Δt+1_P(EQ)][P(EQ) − P(EQ ∩ EQ_+1_)]}(33)
L(1,1,0) = P(1,1,0)/{[1 − G_Δt+1_P(EB)]P(EQ ∩ EQ_+1_)}(34)
L(1,1,1) = P(1,1,1)/[G_Δt+1_P(EB)P(EQ ∩ EQ_+1_)](35)
where the probability gain
G_Δt+1_ = 1 + corr(EQ_+1_,EB_+Δt_){[1/P(EQ) − 1][1/P(EB) − 1]}^½^(36)
is the same as the relation (1) less than for a shift in time of 2 h, G_Δt+1_ is shown in Figure 5.

The probability P(EQ ∩ EQ_+1_) can be estimated by EQ self-correlation using (19); the result coincides with N_EQ ∩ EQ+1_/N_h_ = 8.6 × 10^−4^, where N_EQ ∩ EQ+1_ = 6 is the number of West Pacific EQs followed by other West Pacific EQs in the consecutive 2 h of the same time interval of N_h_ couples of hours. As for G_Δt_(EQ ∩ EQ_+1_), the new histogram Σ_{EQ ∩ EQ+1,EB}_[(EQ ∩ EQ_+1_) *×* EB] was calculated to estimate the correlation in (18). The conditional probability P(EQ ∩ EQ_+1_|EB_+Δt_) = G_Δt_(EQ ∩ EQ_+1_)P(EQ ∩ EQ_+1_) is plotted in Figure 6. This shows only 18 correlation events, which rarely fill the histogram with more than one event per bin. The TE_EB->EQ_ can be calculated by summing the relations (17), (20) to (26), multiplying the relative log_2_ arguments (28) to (35).

However, problems occur when P(EQ ∩ EQ_+1_|EB_+Δt_) is zero, which is a consequence of the low statistic of self-correlated EQs. Really, this conditional probability is always positive. Thus, a factor of 0.4 has been considered for the self-correlated EQs, and the ground value was calculated to obtain the same average value. The new conditional probability of self-correlated EQs has been reported in Figure 7 with respect to the time interval Δt.

Finally, the TE between the lithosphere and the ionosphere is described by the EQs and EBs, respectively. Furthermore, TE_EB->EQ_ based on the new P(EQ ∩ EQ_+1_|EB_+Δt_) is shown in Figure 8.

## 4. Discussion

The uncertainties associated with the variables EQ and EB are very different, where a lower value for EQs, 0.068 bits, is measured with respect to those for EBs, 0.844 bits. The low H(EQ) depends on the EQ = 0 outcome, which is more likely to happen, whereas the higher frequency of EBs is the cause of the near to 1 bit for H(EB). The entropy of the joint system (10) is greater than the singular entropies, as the total number of events increases, and the probability of detecting indifferently an EQ or an EB is closer to 0.5. The joint entropy is lower than the simple sum of each system entropy, as a low non-significant correlation always exists between two sets of uncorrelated measurements, which is by chance. Thus, when the correlation increases between EQs and EBs, H(EQ,EB) decreases. This is evident in Figure 3 when EQs and EBs with a time delay are considered, even if the physical meaning of the joint entropy of a physical system composed of two parts considered at different times is debatable. The same considerations for the physical meaning of the conditional entropy of a physical system composed of two parts are considered at different times. The conditional entropy is the remaining uncertainty regarding EQ when the outcomes of EB are known, and it is at a lower level for Δt of the maximum correlation.

The predominant direction of interaction cannot be directly searched for by mutual information because it is symmetric with respect to EQ and EB [40]; while delayed mutual information is more adapted for analyzing driver-response relationships. Figure 7 shows a significant mutual information relative variation occurring for Δt = 1.5–3.5 h. Although, the maximum value of mutual information is equal to 0.0068 bits, which is 1/10 of the uncertainty H(EQ). Mutual information has the physical meaning of how much information EB has about EQ, and the time delay breaks vice versa. In particular, it means that EB shares a maximum amount of information with the future state of EQ. It was shown that TE, which is based on transition probabilities, allows for better detection of information transfer [40]. Figure 8 reports the TE considering the same time delay intervals Δt of all the previous figures and is in agreement with an excess of information transfer from EB to EQ for Δt = 1.5–3.5 h. Note that TE_EB->EQ_ is not always positive; this is due to the approximation of the conditional probability of self-correlated EQs P(EQ ∩ EQ_+1_|EB_+Δt_) and the contribution of fluctuations. In this regard, it needs to be emphasized that whenever considering 600 EQs during 16.5 years of data, the Cluster 2000 software excluded many EQs in consecutive intervals. This significant reduction in the EQ self-correlation is a direct consequence of the Cluster’s algorithm based on correlations among EQs [34]. However, the presence of a second peak which does not appear in the correlation histogram, nor in either the entropies or the mutual information, seems strictly dependent on the EQ self-correlation. This can be deduced by looking at Figure 6 and Figure 7, where EQ self-correlation shows a peak for the same time delay.

The considered delay Δt = −58.5–−56.5 h represents a real novelty that arises from this analysis. It means that EB shares a maximum amount of information with past EQs occurring around 2 days and 9.5 h before EBs. To better understand this result it is possible to observe that the major contributions to this peak came from relations (17), (21), (22), and (23), where positive terms including P(EQ ∩ EQ_+1_|EB_+Δt_) are summed. In a similar way, the major contributions to the peak with Δt = 1.5–3.5 h came from relations (20) and (25), which sum positive terms containing the correlation (1). Being so, linear and non-linear contributions could be separated, making a distinction between correlation and self-correlation in the particular case of binary events, respectively. For what concerns the possible physical interpretation of the unsuspected TE peak shown in Figure 8, foreshock seismic activity should be investigated. If foreshocks occur, and an average of 2.5 h separate EBs from mainshocks, then foreshocks would anticipate mainshocks of 2 days and 7 h. Thus, the apparent paradox of TE indicating an information flux towards past events can be interpreted as foreshocks that possess information about mainshocks. However, it is necessary to remember that, in this case, self-correlation is obtained with a few EQs, and its significance is very low, even if TE is very sensible for self-correlations.

## 5. Conclusions

In our study here, entropies, delayed mutual information, and TE have for the first time been analytically calculated for their combined system of EQs into the lithosphere and EBs occurring into the ionosphere. This was possible, thanks to previously obtained descriptions of this system by conditional probability [12]. The reported expressions are valid for the general case of each couple of binary event time series belonging to the respective parts of a combined system.

The statistical correlation between EQs in the West Pacific and EBs detected by NOAA resulted in significant variations on entropies of the coupled system, calculated with the same time delays. The delayed mutual information supported, even more, the existence of this correlation. Finally, the TE revealed a consistent transfer of information from EBs to EQs corresponding to the correlation peak, together with a new and less pronounced exchange of information, about two and a half days later.

This last result can be traced back to the occurrence of foreshocks, although this finding appears to be of little significance in this case, where the EQ self-correlation concerns very few cases. Through the analysis of the contributions to the TE, this minor information exchange allows separating the non-linear contribution from the linear one.

## Figures and Tables

**Figure 1 entropy-24-00359-f001:**
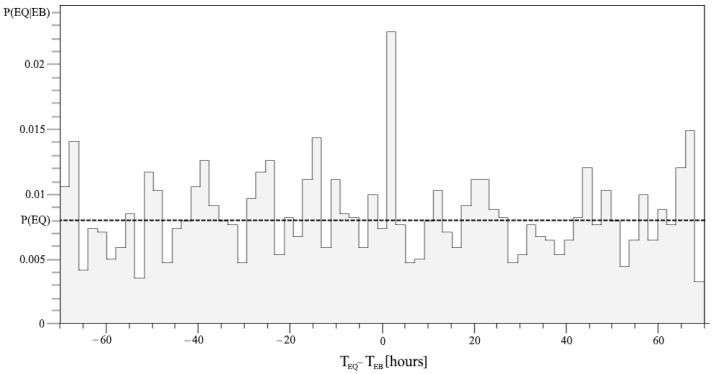
The conditional probability distribution P(EQ|EB) with respect to Δt in the range [–70, 70] hours is defined for each two-hour interval. The dashed line for the average value represents the level of P(EQ).

**Figure 2 entropy-24-00359-f002:**
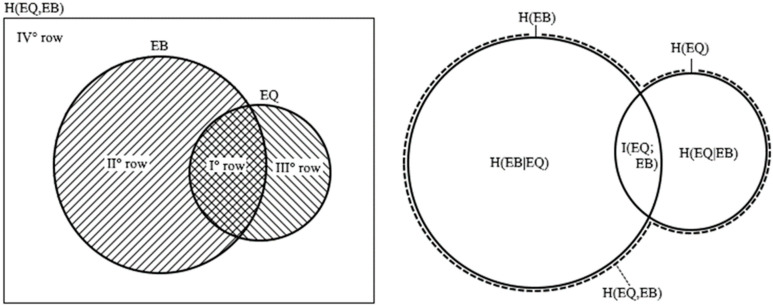
Venn diagrams: for EQ and EB to understand the origin of the rows of the basic Equation (10), on the **left**; for the entropies and the mutual information of EQ and EB events, on the **right**. The set of EBs is drawn larger than the set of EQs because EB events are here more numerous than EQ events.

**Figure 3 entropy-24-00359-f003:**
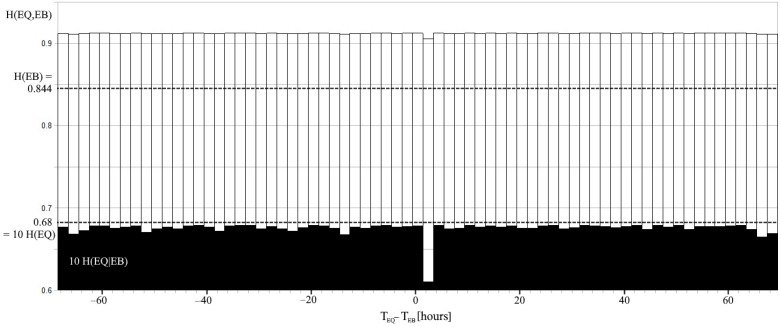
The joint entropy H(EQ,EB) when a delay Δt between EQs and EBs is considered. H(EQ|EB) = H(EQ) and H(EQ,EB) = H(EB) + H(EQ) represent limits of entropies when EQ and EB are independent events. H(EQ) and H(EB) are displayed by dotted lines. Note that H(EQ) and H(EQ|EB) are plotted with a magnification of 10.

**Figure 4 entropy-24-00359-f004:**
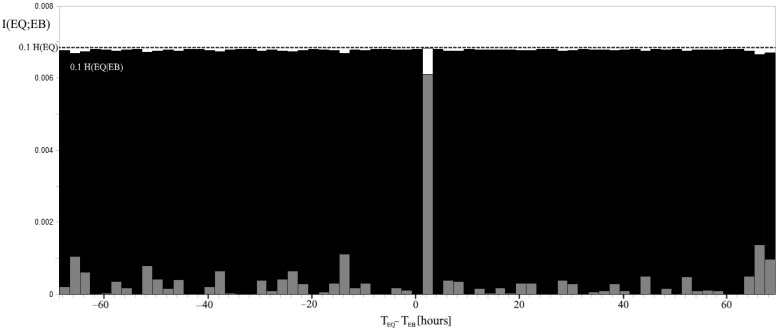
The mutual information I(EQ;EB) obtained by G_Δt_ of [12] when a delay Δt between EQs and EBs is considered in gray. H(EQ|EB) = H(EQ), indicated by a dashed line, represents the limit of the conditional entropy when EQ and EB are independent events. Note that the mutual information increases up to 0.0068 for Δt = 1.5–3.5 h with a relative variation that is greater than that of H(EQ|EB). The H(EQ|EB) level is indicated in gray for Δt = 1.5–3.5 h; H(EQ) and H(EQ|EB) are plotted with an attenuation of 0.1.

**Figure 5 entropy-24-00359-f005:**
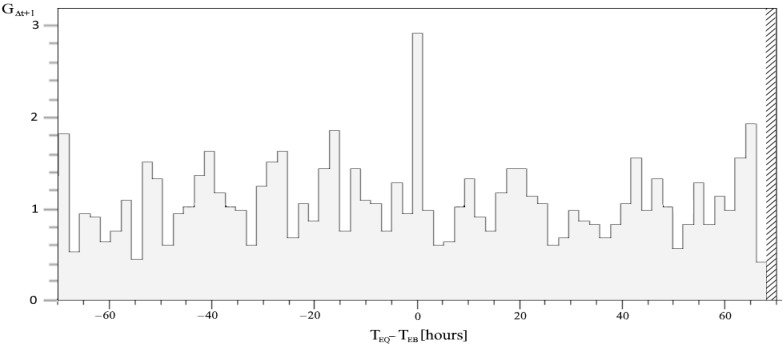
The probability gain G_Δt+1_ is shifted 2 h ahead with respect to G_Δt_. The vertical area of diagonal lines on the right is out of the calculated time difference limits for the correlation.

**Figure 6 entropy-24-00359-f006:**
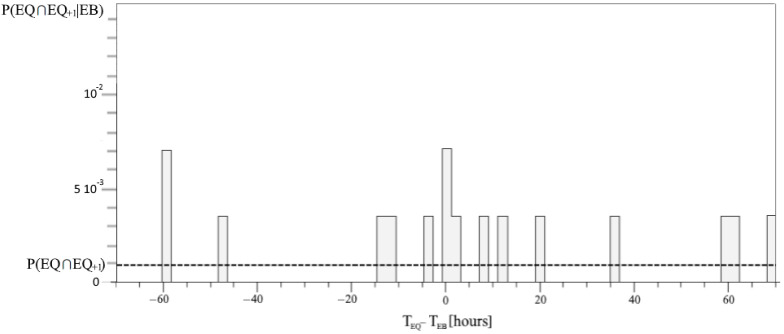
The conditional probability of the self-correlated EQs with respect to the same time interval Δt, and the same time shift, of the correlation in the range [–70, 70] hours; the dashed line of the average value represents the level of P(EQ ∩ EQ_+1_).

**Figure 7 entropy-24-00359-f007:**
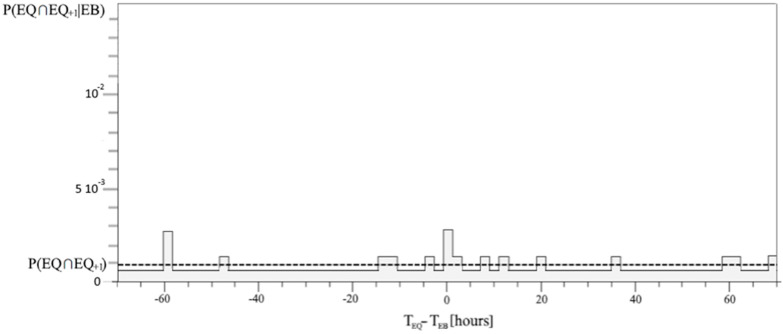
The always positive conditional probability of the self-correlated EQs with respect to the same time interval Δt, and the same time shift, of the correlation in the range [–70, 70] hours; the ground probability has been set based on the same average value of the dashed line.

**Figure 8 entropy-24-00359-f008:**
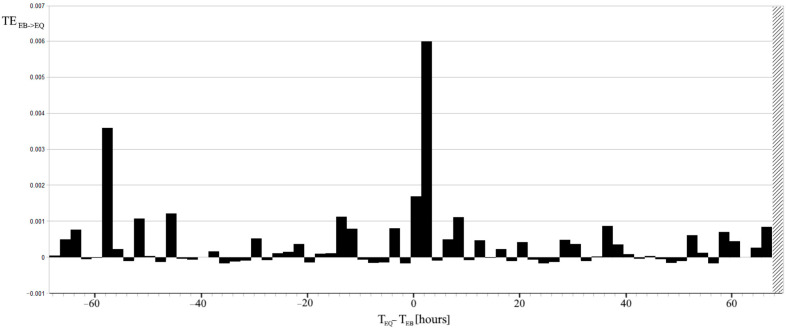
TE between EQs in the lithosphere and EBs in the ionosphere with respect to the time delay Δt, and the same time shift, of the correlation in the range [–70, 70] hours. The vertical area of diagonal lines on the right is out of the calculated time difference limits for the correlation.

## Data Availability

They are reported in Section 2.

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
