# Peer review of "Transfer Entropy of West Pacific Earthquakes to Inner Van Allen Belt Electron Bursts"

_entropy, 2022, doi:10.3390/e24030359_

Round 1

Reviewer 1 Report

It's interesting that this manuscript focuses on transfer entropy of west pacific earthquakes to inner van allen belt electron bursts.

This manuscript proposes the conditional probability statistical model which was deduced from the correlation, the Shannon entropy, the joint entropy and the conditional entropy are calculated.

It is well written and the topic interesting and worth of investigation. I think it can be considered to be accepted.

Author Response

Thanks for comments

Reviewer 2 Report

In this manuscript (ms), the author focuses in the lithosphere-ionosphere coupled complex system and its possible use for the identification of strong earthquake precursors. Using a conditional probability statistical model the author calculates the Shannon entropy, the joint entropy, and the conditional entropy. The method is based on two binary time-series indicating the existence of strong earthquakes and electron burst events evaluated in two-hour intervals (couple of hours). Time delayed mutual information and transfer entropy have been also calculated by including correlations between consecutive earthquake events, and between earthquakes and electron bursts. Peaks have been observed for time delays Δt=1.5-3.5h before strong earthquakes as in the correlation, and a new time delay Δt=(-58.5 -  -58.5) h (i.e., after the strong earthquake) that appears here for the first time. This second peak may be of interest for separating the non-linear contribution of the transfer entropy.

                The paper is well-written, self-explanatory, original, interesting, and presents with the appropriate level of vigor the information entropy concepts used. It improves previous studies by the same author and the conclusions are well supported by the results. I consider the ms certainly deserves publication in Entropy. The following comments may help the author to improve the presentation and readability of the ms:

i)In line 14 the term “digital” events appears. I think that “binary” in general is better suited than “digital” for the purpose of this ms. As a reader, I had to go through many pages of the ms to find what “digital events” mean.

The phrase “A complexity measure … in natural time under time reversal [17]” in lines 76, 77should be extended as follows: “the study of which enables the determination of the occurrence time of major earthquakes in Japan [EPL 130 (2020) 29001] (beyond the estimation of their epicentral regions [Procs. Natl. Acad. Sci USA 112, 986-989 (2015)] when the analysis of seismicity is made in natural time [EPL 91, 59001 (2010); J. Geophys. Res. Space Physics 119, 9192-9206 (2014)]”

ii)In line 165, in the figure caption of Figure 1 the discretization of two-hour intervals should be mentioned for the first time as the definition of G_{Δt} should be given (it appears at a later stage and complicates the reader).

iii)Curly brackets “{}” should be used in some Equations in the denominators and after the square root symbol to eliminate any ambiguity difficulty that a reader may find. The author should take special care in the proof that the Equations are printed in the ms correctly. 

iv)The introduction of a Venn diagram for EQ and EB at l.233 will help the reader understand the origin of the basic Equation (10).

v)After introducing G in lines 240-244, G_{Δt} should be also introduced probably in relation to Figure 1 (see point 2, above). 

Finally, the following typos need correction:

  • The sentence “The non-triviality … non-linear dynamics.” In lines 70-74 needs rephrasing because many gerunds and a verb seems to be missing.
  • In line 202, “(6) a peak” should become “(6) exhibits a peak”
  • In line 204, “Instead”
  • In line 210, “[36].” should become “[36], ”
  • In line 249, “introduced calculating” should become “introduced in calculating”
  • In line 258, “couple of hours” or “pairs of hours” or “two-hour interval”?
  • In line 258, “alarm rate” or “success rate”?
  • In line 266, there is no need to refer to G_{Δt} “of [12]” when it will be clearly defined in the present ms. Please remove reference.
  • In line 294, “transfer entropy can” may become “transfer entropy (TE) can”
  • In lines 322 and 355 “Log_2” should become “log_2”
  • In line 418, “it need be” should become “it needs to be”

In summary, I consider that this ms presents original research in both the methods used and the results found that certainly deserves publication in Entropy. The author should proceed to a minor revision to address the points mentioned above. I will be glad to suggest publication of an appropriately revised version of the ms.

Author Response

i)In line 14 the term “digital” events appears. I think that “binary” in general is

better suited than “digital” for the purpose of this ms. As a reader, I had to go

through many pages of the ms to find what “digital events” mean.

The term “digital” was substituted by “binary” in all the text

The phrase “A complexity measure … in natural time under time reversal [17]” in

lines 76, 77 should be extended as follows: “the study of which enables the

determination of the occurrence time of major earthquakes in Japan [EPL 130 (2020)

29001] (beyond the estimation of their epicentral regions [Procs. Natl. Acad. Sci

USA 112, 986-989 (2015)] when the analysis of seismicity is made in natural time

[EPL 91, 59001 (2010); J. Geophys. Res. Space Physics 119, 9192-9206 (2014)]”

The phrases were added “, the study of which enables the determination of the occurrence time of major EQs in Japan [18]. Furthermore, it enables the estimation of EQ epicentral regions [19] when the analysis of seismicity is made in natural time [20,21].”

and references:

“18 Varotsos, P. A., Sarlis, N. V., and Skordas, E. S. Self-organized criticality and earthquake predictability: A long-standing question in the light of natural time analysis. EPL, 2020, 132, 29001.

19 Sarlis, N. V., Skordas, E. S., Varotsos, P. A., Nagao, T., Kamogawa, M., and Uyeda, S. Spatiotemporal variations of seismicity before major earthquakes in the Japanese area and their relation with the epicentral locations. PNAS, vol. 112. no. 4, 986–989.

20 Sarlis, N. V., Skordas, E. S., and Varotsos, P. A. Order parameter fluctuations of seismicity in natural time before and after mainshocks EPL 91, 59001, 2010.

21 Varotsos, P. A.; Sarlis, N. V.; Skordas, E. S. Study of the temporal correlations in the magnitude time series before major earthquakes in Japan. J. Geophys. Res. Space Phys. 2014, 119, 9192–9206.”

were added. All the other reference numbers were modified accordingly in the text.

ii)In line 165, in the figure caption of Figure 1 the discretization of two-hour

intervals should be mentioned for the first time as the definition of G_{Δt} should

be given (it appears at a later stage and complicates the reader).

“is defined for each two-hour interval” was added in the figure caption of Figure 1; “= GΔt P(EQ)” was added in (1).

iii)Curly brackets “{}” should be used in some Equations in the denominators and

after the square root symbol to eliminate any ambiguity difficulty that a reader

may find. The author should take special care in the proof that the Equations are

printed in the ms correctly.

Yes, the author agrees with the reviewer, and “{}” were used in relations (1), (5), (6), (18), (28), (29), (30), (31), (32), (33), (34), (35), and (36) to eliminate the square root and any ambiguity. All the relations were controlled.

iv)The introduction of a Venn diagram for EQ and EB at l.233 will help the reader

understand the origin of the basic Equation (10).

A Venn diagram of EQ and EB was added in the new Figure 2 left and another Venn diagram of entropy and mutual information interrelations was added in the new Figure 2 right. The Figure 2 caption was written as “Venn diagrams: for EQ and EB to understand the origin of the rows of the basic equation (10), on the left; for the entropies and the mutual information of EQ and EB events, on the right. The set of EBs is drawn larger than the set of EQs because EB events are here more numerous than EQ events.” In the text were added the phrases “The origin of the rows reported in (10) can be clarified in a Venn diagram, see Figure 2 left.” before Figure 2; “Relations among the entropies are represented in Figure 2 right by a Venn diagram.” after the relation (13); and “Relations among the entropies and the mutual information are represented in Figure 2 right by a Venn diagram.” after the relation (16). Figure numbers from 2 to 7 were increased by 1, and the text was modified accordingly.

v)After introducing G in lines 240-244, G_{Δt} should be also introduced probably

in relation to Figure 1 (see point 2, above).

“..., introducing the probability gain GΔt = P(EQ|EB)/P(EQ) by (1) with a time delay Δt = TEQ − TEB of correlations [12] between EQs at the time TEQ and EBs at the time TEB,...” was added after relation (11), “A time delay Δt = TEQ − TEB was introduced calculating correlations [12], where TEQ is the time of the EQ event and TEB is the time of the EB event.” after relation (13) was deleted, and “GΔt” was used in relations (12) and (14).

Figure 3 was coherently modified by filling the white parts of I(EQ;EB) in grey where behind there was H(EQ|EB).

Finally, the following typos need correction:

The sentence “The non-triviality … non-linear dynamics.” In lines 70-74 needs rephrasing because many gerunds and a verb seems to be missing.

It was rewritten as “The non-triviality of the interaction occurs when the system characterized by many components, which if they interacted trivially would be the domain of statistical mechanics, has well-defined and constrained interactions [15], which lead to non-linear dynamics.”

In line 202, “(6) a peak” should become “(6) exhibits a peak” modified

In line 204, “Instead” modified

In line 210, “[36].” should become “[36], ” modified

In line 249, “introduced calculating” should become “introduced in calculating” deleted

In line 258, “couple of hours” or “pairs of hours” or “two-hour interval”? pairs of hours

In line 258, “alarm rate” or “success rate”?

alarm rate. In fact, in Console (2001), it is the rate at which target events are preceded by precursors.

In line 266, there is no need to refer to G_{Δt} “of [12]” when it will be clearly

defined in the present ms. Please remove reference. deleted

In line 294, “transfer entropy can” may become “transfer entropy (TE) can”

transfer entropy was substituted by TE in all the following

In lines 322 and 355 “Log_2” should become “log_2” modified

In line 418, “it need be” should become “it needs to be” modified

Round 2

Reviewer 2 Report

I checked the revised version and found that the author has made all the improvements I recommended except of the fact that in Reference 19 the year of its publication (2015) is missing. After the correction of Ref. 19, I gladly recommend the publication of the present version.